# A Novel Synthesis of Poly(Ester-*Alt*-Selenide)s by Ring-Opening Copolymerization of γ-Selenobutyrolactone and Epoxy Monomer

**DOI:** 10.3390/polym12051203

**Published:** 2020-05-25

**Authors:** Ya’nan Wang, Xiaofang Lin, Zhengbiao Zhang, Jian Zhu, Xiangqiang Pan, Xiulin Zhu

**Affiliations:** 1State and Local Joint Engineering Laboratory for Novel Functional Polymeric Materials, College of Chemistry, Chemical Engineering and Materials Science, Soochow University, Suzhou 215123, China; wangyanan20174@163.com (Y.W.); 20184209201@suda.edu.cn (X.L.); zhangzhengbiao@suda.edu.cn (Z.Z.); xlzhu@suda.edu.cn (X.Z.); 2Jiangsu Key Laboratory of Advanced Functional Polymer Design and Application, College of Chemistry, Chemical Engineering and Materials Science, Soochow University, Suzhou 215123, China

**Keywords:** ring-opening copolymerization, selenide-containing polyester, block polymer

## Abstract

Ring-opening copolymerization (ROCOP) is an effective means to prepare functionalized polyester. In this work, a type of selenide-containing polyesters with controllable structure, molecular weight, and molecular weight distribution was successfully prepared by ROCOP of γ-selenobutyrolactone and epoxy compounds. The influence of the catalyst, solvent, and reaction temperature on the reaction efficiency was examined. Then, kinetic study was investigated under an optimized condition. The structure of the copolymers was carefully characterized by nuclear magnetic resonance (NMR), ^1^H NMR, ^13^C NMR, and ^77^Se NMR, Matrix-assisted laser-desorption-ionization time-of-flight mass spectrometry (MALDI-TOF-MS), and size exclusion chromatography (SEC). The resulting polymers showed a linear structure with a sequence regulated backbone repeating unit of ester-selenide. On this basis, some typical epoxides were investigated to verify the scope of the polymerization system. Due to the “living”/controlled characteristics of this ROCOP, multiblock, amphiphilic, and stereotactic copolymers could be prepared with a pre-designed structure. As expected, the selenide-containing amphiphilic copolymer could self-assemble to micelles and showed an oxidative response.

## 1. Introduction

Selenium is a necessary trace element in living organisms. Since selenium was first discovered, it has caused widespread concern [1]. From the synthetic application of selenium-containing compounds to the synthesis of selenium-containing polymers by various polymerization methods, after years of exploration and development, the foundation for the application of selenium-containing polymers has been laid [2,3,4,5,6]. The stimuli-responsive selenium-containing polymers have attracted a great deal of attention in recent years and have been widely used as biomedical materials and adaptive materials [7,8,9]. The relationship between selenium content and photoelectric properties of polymeric materials has also been investigated in the past decades [10,11,12]. In order to obtain more diverse selenium-containing polymers for a property study, the development of synthetic methods should be given attention to continuously. The selenide-containing polymer is a kind of typical selenium-containing polymer, which can be prepared by the following methods: stepwise polymerization, controlled radical polymerization [13,14], and ring opening polymerization (ROP). Stepwise polymerization has been developed to prepare selenide-containing polymers for several decades. Selenide-containing polymers could be prepared with poor solubility and poor structure designability [15]. In recent years, Xu et al. improved this strategy and designed a new type of selenide-containing polymer for bio-application by introducing a flexible or hydrophilic segment into the polymer main chain [7]. Organoselenium mediated controlled radical polymerization was found to be another strategy for the preparation of selenide-labeled polymers [16,17]. Normally, selenide moieties are introduced at the end of the polymers. ROP is also a good prospect for the preparation of polymers with a high selenium content. Lang et al. synthesized selenium-containing aliphatic polycarbonates via lipase-catalyzed ring-opening homopolymerization of a selenic macrocyclic carbonate monomer [18]. Li et al. reported the ROP of Se-functional lactone and cyclic carbonate monomers to give biodegradable and oxidation-responsive polymers [19,20]. 

Copolymerization is an effective strategy for designing polymers with regulable properties by incorporating two or more different monomers into one polymer chain and controlling the composition and sequence structure of the copolymers. γ-Selenobutyrolactone was found to be a new synthon, which has been developed in our group for constructing diselenide-containing polymers in the last few years [21,22]. Its five-membered ring structure provides a chance for aminolysis and hydrolysis to construct diselenide-containing polyamides and polyesters. However, ring-opening (co)polymerization of γ-selenobutyrolactone has yet to be well implemented. Up until now, there has been only one work—reported from the Zhang group—on the copolymerization of γ-selenobutyrolactone and epoxides mediating the phosphazene base/alcohol [23]. That was a typical living anionic copolymerization to give copolymers with an alternating structure. Herein, we report on a new catalytic system for ring-opening copolymerization of γ-selenobutyrolactone with epoxides. Selenide-containing polyesters with a controllable structure were obtained through metal-free ring-opening copolymerization by using quaternary ammonium salt (Scheme 1). No any other initiator was needed in our strategy and the catalyst was commercially available and cheap. Furthermore, amphiphilic and stereotactic selenide-containing copolymers were first prepared by using this strategy. 

## 2. Materials and Methods 

### 2.1. Materials 

Tetrabutylammonium bromide (TBAB, 99%) and tetrabutylammonium chloride (TBAC, 98%) were purchased from 3A (Shanghai, China) Chemicals Co. Ltd. and recrystallized twice from ethyl acetate. Tetrabutylphosphonium bromide (TBPB, 99%) and glycidyl 4-methoxyphenyl ether (MPE, 98%) were purchased from Alfa Aesar (Shanghai, China) Chemical Co. Ltd. Tetrabutylphosphonium chloride (TBPC, 98%), allyl glycidyl ether (AGE, 98%), and styrene oxide (SO, 98%) were purchased from Adamas Reagent Co. Ltd (Shanghai, China). The 18-crown-6 (18-C-6, 98%) and potassium chloride (KCl, AR) were purchased from Sinopharm Group Chemical Reagent Co. Ltd (Shanghai, China). Glycidyl phenyl ether (GPE, 98%), butyl glycidyl ether (BGE, 98%), cyclohexene oxide (CHO, 98%), and glycidyl propargyl ether (BEA, 90%) were purchased from TCI (Shanghai, China) Development Co. Ltd. (S)-(+)-glycidyl benzyl ether(*S*-DGA, 98%), (R)-(−)-glycidy benzyl ether(*R*-DGA, 98%) were purchased from Meryer (Shanghai, China) Chemical Technology Co. Ltd. Ethyl acetate (AR), chloroform (AR), anisole (AR), toluene (AR), hexane (AR), and dimethyl sulfoxice (DMSO, AR) were purchased from Yonghua (Jiangsu, China) Technology Co. Ltd. N,N-dimethylformamide (DMF, AR) was purchased from Sinopharm Group Chemical Reagent Co. Ltd. All of the chemicals above-mentioned were used as received without further purification. γ-selenobutyrolactone (SBL) was prepared according to [24]. 

### 2.2. Measurement 

^1^H NMR, ^13^C NMR, and ^77^Se NMR spectra were obtained on a Bruker Avance 300 at 300 MHz or on a Bruker Avance 400 at 600 MHz (CA, USA). CDCl_3_ was used as the solvent. Chemical shifts of CDCl_3_ (7.26 ppm in ^1^H and 77.00 ppm in ^13^C NMR, respectively) or diphenyl selenide (416.00 ppm) were obtained in ^77^Se NMR. Coupling constants (J) in ^1^H NMR are given in Hz. S (single), d (doublet), t (triplet), q (quartets), or m (multiplet) describe the resonance multiplicities. MALDI-TOF -MS was carried out on a beam-II laser with a 355 nm wavelength and 200 Hz firing rate. Trans-2-[3-(4-tert-butyl-phenyl)-2-methyl-2-propenylidene]-malononitrile (DCTB, Aldrich, Shanghai, China >98%) was dissolved in CHCl_3_ at a concentration of 20 mg mL^−1^ and used as the matrix. The cationizing agent sodium trifluoroacetate was dissolved in ethanol at a concentration of 10 mg mL^−1^. The solutions were mixed in a ratio of 10/1 (*v/v*). The polymer was dissolved in CHCl_3_ at a concentration of 10 mg mL^−1^. The mixed solution and polymer solution were alternately dropped on the target plate. The data were further processed using Bruker Daltonics flexAnalysis Software. SEC (Tokyo, Japan) as performed using TOSOH HLC-8320 gel permeation chromatograph, and THF was used as the eluent with 0.35 mL min^−1^ at 40 °C. Calibration used were the polystyrene or poly(methyl methacrylate) standards. Thermogravimetric analysis (TGA) was measured by a TG/DTA 6300 Instruments (MA, USA) with a heating rate of 10 °C min^−1^ under a nitrogen atmosphere. Differential scanning calorimetric analysis (DSC) was performed using a TA Instruments Discovery Q200 (DE, USA) with a heating rate of 10 °C min^−1^ under a nitrogen flow of 20 mL min^−1^. The CD spectra were performed using a JASCO J-815 (Tokyo, Japan) spectropolarimeter equipped with a Peltier-controlled housing unit using a SQ-grade cuvette. UV−Vis spectra were carried out on a Shimadzu UV-3150 spectrophotometer (Shanghai, China). Dynamic light scattering (DLS) was measured by a Brookhaven NanoBrook 90Plus PALS (New York, NY, USA) instrument with a scattering angle of 90° at 25 °C. The Transmission electron microscope (TEM) were taken using the FEI TecnaiG220 (120 KV) (OR, USA). 

### 2.3. Typical Polymerization Procedures

Copolymerization of **SBL** with **GPE**. **GPE** (0.15 g, 1.0 mmol), **SBL** (0.15 g, 1.0 mmol), and TBAB (0.006 g, 0.02 mmol) were all charged into a glass tube. The mixture was heated to 80 °C for 24 h under stirring in an argon atmosphere. After that, the reaction mixture was dissolved with 2 mL of chloroform, and then poured into hexane to precipitate the polymer. The resulting polymer was reprecipitated twice from chloroform into hexane and dried in vacuum. The yield of the corresponding poly(ester-*alt*-selenilde) (**PSe-1**) was 0.30 g (98.5%). The number average molecular weight (*M*_n_) of the polymer determined from SEC was 8100 g mol^−1^ (*Ð* = 1.44). ^1^H NMR (300 MHz, CDCl_3_, TMS): δ 1.88–2.04 (m, 2H, SeCH_2_C**H**_2_), 2.36–2.54 (m, 2H, CH_2_C**H**_2_CO), 2.54–2.71 (m, 2H, SeC**H**_2_CH_2_), 2.71–2.96 (m, 2H, CO_2_CHC**H**_2_), 4.08–4.21 (m, 2H, CHC**H**_2_OPh), 5.14–5.31 (m, 1H, CO_2_C**H**CH_2_), and 6.81–7.34 (m, 5H, Ar**H**). ^13^C NMR (75 MHz, CDCl_3_, TMS): δ 23.51, 23.95, 25.45, 33.74, 67.99, 71.94, 114.65, 121.28, 129.29, 157.91, 172.27.

Preparation of poly(**SBL**-*alt*-**GPE**)-*b-*poly(**SBL**-*alt*-**TGE**). **SBL** (0.30 g, 2.0 mmol), **GPE** (0.15 g, 1.0 mmol), and TBAB (0.012 g, 0.04 mmol) were all charged into a 10 mL Schlenk flask. The mixture was heated to 80 °C for 12 h under stirring in an argon atmosphere. Then, **TGE** (0.22 g, 1.0 mmol) was injected to the reaction solution and kept for another 24 h at 80 °C. The reaction mixture was dissolved with 2 mL of chloroform, and then poured into hexane to precipitate the polymer. The resulting polymer was reprecipitated twice from chloroform into hexane and dried in vacuum. The number average molecular weight (*M*_n_) of the polymer determined from SEC was 7400 g mol^−1^ (*Ð* = 1.49) ^1^H NMR (300 MHz, CDCl_3_, TMS): δ 1.88–2.06 (m, 4H, SeCH_2_C**H**_2_), 2.34–2.47 (t, 4H, CH_2_C**H**_2_CO), 2.55–2.70 (t, 2H, SeC**H**_2_CH_2_), 2.72–2.97 (m, 2H, CO_2_CHC**H**_2_), 3.38 (s, 3H, CH_2_OC**H**_3_), 3.50–3.70 (d, 12H, OCH_2_C**H**_2_), 4.08–4.19 (d, 4H, CHC**H**_2_O), 5.14–5.32 (m, 2H, CO_2_C**H**CH_2_), and 6.82–7.33 (m, 5H, Ar**H**).

Polymeric micelles self-assembled from amphiphilic polymers. Poly(**SBL**-*alt*-**GPE**)-*b*-poly(**SBL**-*alt*-**TGE**) was dissolved in THF with a concentration of 2 mg mL^−1^, then water was added dropwise in 0.2 mL h^−1^ and the mixture was stirred for 3 h. After that, the resulting solution was put into a dialysis bag for dialysis against distilled water for two days to remove the THF.

## 3. Results and Discussion

### 3.1. Copolymerization of SBL and GPE 

A systematic survey of reaction conditions varying the quaternary ammonium salt, the solvent, and the temperature was carried out, and the most relevant results are summarized in Table 1 and Appendix A. Here, γ-selenobutyrolactone (**SBL**) or glycidyl phenyl ether (**GPE**) was treated with TBAB for 24 h, respectively (Scheme 2). No polymer was detected indicating that homopolymerization of the two monomers did not occur in the current condition. Mixing **SBL** and **GPE** with a molar ratio of 1:1 without a catalyst, the polymer was also not detected after 24 h at 80 °C (Table 1, Entry 1). In addition, polymers were obtained when several quaternary ammonium salts were added into the reaction solution, respectively. Among these catalysts, tetrabutylammonium bromide (TBAB) worked best, giving a polymer with high molecular weight and narrow molecular weight distribution (Table 1, Entry 5). Some typical solvents were used in this polymerization using 5 mol% of TBAB as the catalyst at 80 °C. Yields of polymer were 60, 74, 98, 98, and 98% in DMSO, DMF, anisole, toluene, hexane for 24 h, respectively. This indicated that the copolymerization of **SBL** with **GPE** proceeded smoothly in bulk and nonpolar solvents. During the whole reaction processing, the Se^-^ and O^-^ acted as nucleophilic species. Then, they moved on with the ring-opening reaction according to bimolecular nucleophilic substitution mechanism (SN_2_). It is generally known that nucleophilic species show higher activity in nonpolar solvents than in polar solvents due to the weak solvation effect [25]. The effect of the reaction temperature on the copolymerization was examined without solvent for 24 h, in which 5 mol% TBAB was used as the catalyst. As shown in Appendix A, the product yield increased with increasing temperature. The molecular weight of the polymer increased with increasing temperature from 20 to 80 °C, and decreased when the temperature was up to 100 °C. This is because the probability of side reactions such as ester exchange and back-biting degradation increased with increasing reaction temperature. These side reactions frequently occur in the ring-opening polymerization of lactones. The amount of catalyst was also investigated. The molecular weight of polymers was not decreased linearly with the increasing in catalyst proportion (Appendix A). In the kinetic study of this polymerization, the results showed that the molecular weight increased linearly with the increase in monomer conversion and the molecular weight distribution was narrow. These results indicate ‘‘living’’ characteristics of this ROCOP (Appendix A). In the later period of polymerization, the molecular weight increased rapidly and the molecular weight distribution became broader. This is because of the nucleophilic attack of the terminal selenolate anion to the bromine group at another polymer chain toward the end of the reaction. 

As shown in Figure 1 and Appendix A, the structure of the polymer was carefully characterized by the ^1^H NMR, ^13^C NMR, and ^77^Se NMR spectra. In the ^1^H NMR spectrum, the proton signals of –SeCH_2_CH_2_CH_2_– in the **SBL** unit appeared at 2.63 ppm (*I*_2.54–2.71_ = 2.00) (a), 1.96 ppm (*I*_1.88–2.04_ = 2.05) (b), and 2.45 ppm (*I*_2.36–2.54_ = 2.00) (c), respectively. The signal at 5.23 ppm (*I*_5.14–5.31_ = 0.95) (e) was ascribed to the methine protons of the **GPE** unit in the main chain; the signal at 2.84 ppm (*I*_2.71–2.96_ = 2.00) (f) was ascribed to the methylene protons of **GPE** unit in the main chain; and the signal at 4.15 ppm (*I*_4.08–4.21_ = 2.03) (g) was ascribed to the methylene protons of the **GPE** unit in the side chain; and the signal at around 6.81–7.34 ppm (*I*_6.81–7.34_ = 5.73) (h) was ascribed to the aromatic protons of **GPE**. The proton signals of the polymer end group such as **GPE** and tetrabutylammonium were also found in the ^1^H NMR spectrum. The intensity ratio of the methine protons (e) vs. aromatic protons (h) indicated no evidence of the homopolymer of **GPE**. The intensity ratio of the methylene protons (a) vs. aromatic protons (h) indicated an alternating structure of the copolymer. The ^13^C NMR spectrum also showed corresponding signals of the copolymer (Appendix A). The ^77^Se NMR spectrum only showed a single peak at 130.07 ppm according to the selenide in the polymer main chain. In addition, MALDI-TOF-MS was used to further characterize the structure of the copolymer. As shown in Figure 2, the main population of the isotropic peak at 5872.88 m/z matched the theoretical calculation well [(GPE + SBL)_18_ + GPE + TBAB + H^+^, 5872.75 m/z]. Furthermore, two main sequence peaks were very close to the unit of **GPE** and **SBL** (300.03 Da). All of the evidence confirmed that the new copolymer (**PSe-1**) was obtained by the ring-opening alternation copolymerization of **GPE** with **SBL** using TBAB as the catalyst.

Based on the results, the polymerization mechanism was proposed in Scheme 3. Similar to the copolymerization of epoxide with thiolactone, TBAB is composited with epoxide in the initial step [26]. The complex cannot initiate the homopolymerization of epoxide, but opens the five-member ring of γ-selenobutyrolactone to produce ester labeling a selenolate anion. The chain propagation step in this polymerization is repetitions of the addition of epoxide and **SBL** to each propagating species, respectively. A strictly alternating copolymer could be obtained regardless of the ratio of the monomer feeds. In this condition, SBL and epoxide had a same conversion rate (Appendix A), and the molecular weight of the polymer decreased with the increase of another monomer ratio (Appendix A). When the epoxide was exhausted in the later stage of polymerization, the molecular weight of the polymer continued to grow due to the step growth between the polymer chains.

### 3.2. Available Scope of Epoxides

In order to explore the scope of this copolymerization for further application, some typical epoxides such as **MPE**, **BGE**, **SO**, **AGE**, **BEA**, and **CHO** were selected as functional monomers (Scheme 4). As summarized in Table 2 and Appendix A, copolymerization of **SBL** with most of the glycidol derivatives proceeded to give corresponding copolymers with high yields, respectively. Glycidol derivatives have a low price and wide source. This gave a chance for modification using “click chemistry” when the alkenyl and alkynyl group were introduced to the side chain of this type of selenide-containing polymer. The copolymerization of **SBL** with **SO** produced the corresponding polymer. The selectivity of the β-cleavage of **SO** was 76% because of the high electrophilicity of α-carbon on the **SO** comparison with that of the glycidol derivatives. In another case, due to the steric hindrance effect, the copolymerization of **SBL** with **CHO** did not proceed to produce any polymer. Furthermore, chiral **DGA** was used to polymerize with **SBL** (Scheme 4). Poly(**SBL**-*alt*-***R***-**DGA**) and poly(**SBL**-*alt*-***S*-DGA**) showed a positive and negative Cotton effect, respectively (Appendix A). The obvious circular dichroism (CD) signals indicated that the chiral was maintained during the polymerization [27].

### 3.3. Preparing of Block Copolymers

According to the copolymerization mechanism and reaction kinetics, it is possible to design and prepare a multiblock copolymer (Scheme 5). For example, with the [**SBL**]_0_/[**TBAB**]_0_ feed ratio of 50/1, 0.2 equiv. **MPE** was first added to the system and was completely consumed at 6 h, while the residual **SBL** was not homopolymerized. Then, 0.2 equiv **GPE** and **MPE** were introduced into the reaction solution alternately, until **SBL** was almost completely consumed. As shown in Figure 3, a pentablock copolymer with unimodal SEC curve and narrow molecular weight distribution was obtained, though a tailing phenomenon was observed in the low molecular weight region due to some side reaction as above-mentioned.

We also successfully prepared diblock copolymer poly(**SBL**-*alt*-**GPE**)-*b*-poly(**SBL**-*alt*-**MPE**) through the feeding of **GPE** and **MPE** into the **SBL** solution alternately and random copolymer poly(**SBL**-*alt*-**GPE**)-*r*- poly(**SBL**-*alt*-**MPE**) by one pot feeding of **GPE** and **MPE** into the **SBL** solution (Appendix A). Thermal analysis of these copolymers was investigated carefully. As shown in Figure 4, the glass transition temperature (*T*_g_) of poly(**SBL**-*alt*-**MPE**) was higher than that of poly(**SBL**-*alt*-**GPE**). *T*_g_ of poly(**SBL**-*alt*-**GPE**)-*r*-poly(**SBL**-*alt*-**MPE**) was in between, which indicated a selenide-containing random copolymer was prepared successfully. Additionally, only one *T*_g_ of poly(**SBL**-*alt*-**GPE**)-*b*-poly(**SBL**-*alt*-**MPE**) was found to be a little higher than that of poly(**SBL**-*alt*-**GPE**)-*r*-poly(**SBL**-*alt*-**MPE**) due to the short polymer chain and there was little difference of the copolymers poly(**SBL**-*alt*-**MPE**) and poly(**SBL**-*alt*-**GPE**).

### 3.4. Self-Assembly of Amphiphilic Copolymer

To the best to our knowledge, selenide-containing amphiphilic polymers have shown good development and application prospects for controlled drug delivery systems. Encouraged by the above results, a new type of selenide-containing amphiphilic polyester was prepared successfully by using **GPE** and **TGE** as a monomer in turn. Polymeric micelles self-assembled from this polymer (*M*_n_,_SEC_ = 7400 g mol^−1^, *Ð* = 1.49). 

The morphology and diameter were characterized by dynamic light scattering (DLS) and transmission electron microscopy (TEM). As shown in Figure 5, the average diameter of these micelles was 45 nm and 40 nm, respectively. When hydrogen peroxide was added into the micelle solution with a final concentration of 0.1 M, we could observe that the turbid micelle solution eventually became clear [28,29]. This morphology change implied the cleavage of selenide bonds in the polymer and the disassembly of micelles (Appendix A). The results of the SEC also proved that the polymers were degraded. All of these results indicated that the diblock amphiphilic copolymer was synthesized successfully and showed oxidative response.

## 4. Conclusions

In summary, we developed a new method for the synthesis of selenide-containing polyesters. All of the epoxides are commercially available or could be prepared easily from glycidol. The “living” characteristics of this polymerization gave us a chance to design and prepare selenide-containing multiblock copolymers with a variable structure such as stereoregular copolymers, amphiphilic copolymers, and side chain functionalized copolymers. Furthermore, the obtained polyesters displayed an increased biodegradability sensitivity to oxidation. Their further application in many responsive materials could therefore be predicted.

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
