# Peer review of "A Novel Synthesis of Poly(Ester-Alt-Selenide)s by Ring-Opening Copolymerization of γ-Selenobutyrolactone and Epoxy Monomer"

_polymers, 2020, doi:10.3390/polym12051203_

Round 1
Reviewer 1 Report
The manuscript concerns coROP of epoxides with γ-butyroselenolactone resulting in novel Se-containing copolymers that are prospective due to ability to form micelles and oxidative response. The presence of Se-containing subunits expands the possibilities of analyzing copolymer microstructure by 77Se NMR. The article is of interest for the readers of Polymers and can be accepted.
So, I recommend major revision, in which the authors should re-organize the manuscript to improve the quality of presentation.
- The part of the Introduction (lines 42-55) is fragmented and incoherent. I strongly recommend to draw new Scheme to present different approaches to Se-containing polymers, with emphasizing the efficiency of the ROP approach. The key structure of the Se-comonomer, γ-butyroselenolactone, should be presented in this Scheme, precisely in the Introduction section.
- Please re-write Section 2.1 to point manufacturers of the chemicals. The protocol (or reference) for γ-butyroselenolactone is needed.
- Please subdivide the Section 3 as, for example: 3.1. Synthesis SBL/GPE etc; 3.2. Microstructure SBL/GPE etc; 3.3. SBL/epoxide copolymers etc; 3.4. Micelles, oxidative response etc
- Please make up your mind with the terminology: living or chain-growth? Lines 159-166: please, make clear your interpretation
- Technical remarks:
a huge number of errors related to the sequence of tenses, poor quality of the text in general
γ-butyroselenolactone (SBL): maybe γ-selenobutyrolactone (SBL)?
the references in the manuscript should be cited as [XX] (not as .[XX])
line 108: replace Polymerizatin by Polymerization
Author Response
Comments to the Author Q1: The part of the Introduction (lines 42-55) is fragmented and incoherent. I strongly recommend to draw new Scheme to present different approaches to Se-containing polymers, with emphasizing the efficiency of the ROP approach. The key structure of the Se-comonomer, γ-butyroselenolactone, should be presented in this Scheme, precisely in the Introduction section. A1: Thanks very much for your professional comment. The introduction was rewrote to make it more logical. A new Scheme including structure of Se-monomer was also added below the introduction for emphasizing the new ring-opening copolymerization approach. Q2: Please re-write Section 2.1 to point manufacturers of the chemicals. The protocol (or reference) for γ-butyroselenolactone is needed. A2: Thanks very much for your professional comment. Section 2.1 was rewrote, and the protocol for γ-selenobutyrolactone was also supplied. Q3: Please subdivide the Section 3 as, for example: 3.1. Synthesis SBL/GPE etc; 3.2. Microstructure SBL/GPE etc; 3.3. SBL/epoxide copolymers etc; 3.4. Micelles, oxidative response etc. A3: Thanks very much for your professional comment. The Section 3 was subdivided to make the discussion clearly. Q4: Please make up your mind with the terminology: living or chain-growth? Lines 159-166: please, make clear your interpretation A4: Thanks very much for your professional comment. This ROCOP of SBL and epoxide had the characteristics of “living” polymerization. The nucleophilic attack of terminal selenolate anion to bromine group at another end of polymer chain will take over the polymerization toward the end of reaction due to exhaustion of epoxide. The interpretation was corrected to avoid misunderstanding. Q5: Technical remarks: a huge number of errors related to the sequence of tenses, poor quality of the text in general γ-butyroselenolactone (SBL): maybe γ-selenobutyrolactone (SBL)? the references in the manuscript should be cited as [XX] (not as .[XX]) line 108: replace Polymerizatin by Polymerization A5: Thank you for your critical comment to further improve the quality of the manuscript. The whole manuscript was carefully revised. γ-Selenobutyrolactone was used in this manuscript. All the mentioned mistakes were corrected.
Reviewer 2 Report
Pan and Zhu present the ROCOP of butyroselenolactone and epoxy Monomers. The results is useful for the researcher of new material design and synthesis. Publication after the revision is suggested.
- In page 4, line 150-151, the auther explain the nonpolar solvent reveal greater polymerization yield according the nucleophilic mechanism. In order to let the reader know the meaning of the author clearly, this explanation should be modified to be more detailed and clear.
- In page 4, the author mentioned that “γ-butyroselenolactone (SBL) and glycidyl phenyl ether (GPE) were treated with TBAB for 24 h, respectively. No polymer was detected which indicating homopolymerization of the two monomers did not occur in current condition.”. However, the mixture of two monomers can be polymerized by TBAB. I know the similar results of ROCOP of γ-thiobutyrolactone and oxirane were reported before, but can the author explain this situation?
- To explain the table S1 and Fig S2, the author described that “This is because that nucleophilic attack of terminal selenolate anion to another end-group inducing a chain-growth polymerization in the ring-opening polymerization system”. The author should draw the situation in the figure. My question is that when TBAB loading increased, why the polymer mass decreased? Why was the polymer mass decreased linearly with an increased TBAB? Why was nucleophilic attack of terminal selenolate anion to another end-group influenced by TBAB loading? The author should explain it clearly.
- In Page 4, line 155-156, when polymerization temperature was set at 100 oC, the polymer mass decreased, and the author explained it because of some side reaction increases with increasing reaction temperature, such as ester exchange and back-biting degradation. If it is true, the polydispersity should be broad. I suggest the author should provide the polydispersity data of all polymers in Fig S1 to prove it.
- In Fig 1, what is the peak at 410 ppm in 77Se NMR spectrum? Is it a standard? What is the standard? In addition, the peak f revealed a doublet of doublets peak, and I think it was influenced by the selectivity of peak e carbon. But the peak g did not reveal a doublet of doublets Do the author have any expansion?
Author Response
Q1: In page 4, line 150-151, the author explain the nonpolar solvent reveal greater polymerization yield according the nucleophilic mechanism. In order to let the reader know the meaning of the author clearly, this explanation should be modified to be more detailed and clear.
A1: Thank you for your critical comment to further improve the quality of the manuscript. This explanation was modified to be more detailed and clear.
During the whole reaction processing, the Se- and O- acted as nucleophilic species. Then they got on with opening ring reactions according to bimolecular nucleophilic substitution mechanism (SN2). It is generally known that nucleophilic species shown higher activity in nonpolar solvents than in polar solvents due to the weak solvation effects.
Q2: In page 4, the author mentioned “γ-butyroselenolactone (SBL) and glycidyl phenyl ether (GPE) were treated with TBAB for 24 h, respectively. No polymer was detected which indicating homopolymerization of the two monomers did not occur in current condition.”. However, the mixture of two monomers can be polymerized by TBAB. I know the similar results of ROCOP of γ-thiobutyrolactone and oxirane were reported before, but can the author explain this situation?
A2: Yes, the work was reported before and cited in our work (ref 26: Nishikubo, T.; Kameyama, A.; Kawakami, S., A novel synthesis of poly(ester-alt-sulfide)s by the ring-opening alternating copolymerization of oxiranes with gamma-thiobutyrolactone using quaternary onium salts or crown ether complexes as catalysts. Macromolecules 1998, 31, 4746-4752.). That work was only focus on ROCOP behavior study of γ-thiobutyrolactone and oxirane. Our work not noly studied polymerization behavior but also structure design. Multi-stimuli responsive polymers have been a hot topic in recent years due to its application in smart materials. A new type of selenide-containing polyester could be prepared using our strategy. It has a good development prosprect based on the responsiveness and biodegradability of this type polymers.
Q3: To explain the table S1 and Fig S2, the author described that “This is because that nucleophilic attack of terminal selenolate anion to another end-group inducing a chain-growth polymerization in the ring-opening polymerization system”. The author should draw the situation in the figure. My question is that when TBAB loading increased, why the polymer mass decreased? Why was the polymer mass decreased linearly with an increased TBAB? Why was nucleophilic attack of terminal selenolate anion to another end-group influenced by TBAB loading? The author should explain it clearly.
A3: Thanks very much for your professional comment. This explanation was modified to be more detailed and clear in the main text. The nucleophilic attack of terminal selenolate anion to bromine group at another end of polymer chain will take over the polymerization toward the end of reaction due to exhaustion of epoxide. As shown in the proposed mechanism, TBAB is composited with epoxy group in the initial step to generate initiating active species. Subsequently, these active species initiated the “living” ring-opening copolymerization. Therefore, the molar ratios of monomers and the initiator was decreased with an increased TBAB due to decreasing of molecular weight of the polymer. However, the molecular weight of the polymer did not decrease with increasing of catalyst proportion. It indicated that not all TBAB were composited with epoxy group to generate initiating active species and the initiating efficiency was unable to reach 100%.
Q4: In Page 4, line 155-156, when polymerization temperature was set at 100 oC, the polymer mass decreased, and the author explained it because of some side reaction increases with increasing reaction temperature, such as ester exchange and back-biting degradation. If it is true, the polydispersity should be broad. I suggest the author should provide the polydispersity data of all polymers in Fig S1 to prove it.
A4: Thanks very much for your professional comment. The molecular weight distribution data of all polymers in Fig S1 was provide in Table S1 (entries 2-6). As shown in Figure S1, SEC traces of these polymers became broad with rising polymerization temperature. However, the molecular weight distribution was broad and did not change significantly due to nucleophilic attack of terminal selenolate anion to bromine group at another end of polymer chain toward the end of reaction. The step-growth process had great contribution to the molecular weight distribution.
Q5: In Fig 1, what is the peak at 410 ppm in 77Se NMR spectrum? Is it a standard? What is the standard? In addition, the peak f revealed a doublet of doublets peak, and I think it was influenced by the selectivity of peak e carbon. But the peak g did not reveal a doublet of doublets Do the author have any expansion?
A5: Yes, diphenyl selenide was used as a standard for 77Se NMR measurement. The signal at 416 ppm is ascribed to the selenium of diphenyl selenide. In theory, the peak f and g should reveal as a doublet of doublets peak. I think the obvious difference was due to the position of the two -CH2- group. In the polymer main chain, the chain movement is limited. The chemical environment of two protons at methylene group was difference resulting a doublet of doublets peak. By contrast, the movement of polymer side chain is free due to convergence of the two protons at methylene group. Therefore, the peak g revealed a doublet peak influencing by the proton of methine proton.

Reviewer 3 Report
You have well presented a Ring-Opening Copolymerization of γ-3 Butyroselenolactone and Epoxy Monomer; usign a variety of epoxidic derivates and supporting your thesis with analysis and data.
My main concern is more about the way you communicate the results, the aim of a paper is being clear, not interpretable; the results you show have to arrive to the reader unambiguously.
You have a lot of data and you did all the analysis neccessary to be pubblished, but there are errors that make me ask myself if you have done it on purpose: i'd like to give you this main take home message.
My comments are (from the beginning):
The abstract has to be clear:
-line 17-19
The influence of the catalyst, solvent, and reaction temperature on the reaction efficiency was examined. And then kinetic study was investigated under optimized condition.
"And" at the beginning is an error, a full stop is different from a semicolon
-line 21
you meant MALDI?
-line 54-55
you said:<<However, these monomers were needed to prepare by specific reaction process and it is difficulty to derivatization of these monomer to given more possibilities for polymer modification.>> I really don't know what you want to say...the preposition "to" has to be borne by a subject and a verb.
-line 69
you want to say that Zhang et Al. did something similar, but you end up by pasting a phrase at the end of your introduction: it sounds weird. Moreover is it the same person as one of the coauthor of the paper? if yes, said and reshape the frase, why you wanted to mention that?
-line 88-89
add some information more about the MALDI: the matrix you used, the power of the laser etc., how you analized, practially, the polymers, it is important if someone want to replicate the analysis. If you were inspired by a pubblished protocol cite it.
-line 99-100
in the same phrase cm-1 is appearing in two different forms: being homogeneous.
-line 108
you miss a letter in the title
-Scheme 1
you have to cite what is "Cat." up to the first arrow. The reader should be able to understand the scheme without reading the text.
-line 172
MALDI-TOF you already declare the acronym...but MS no!, you have to declare all the acronym (even if obvious).
-line 186
I'm not an expert in a MALDI, but as is not a HR-MS I would use less decimals
Resolution and mass accuracy in matrix-assisted laser desorption ionization- time-of-flight https://doi.org/10.1016/S1044-0305(98)00069-5
-Figure 1
the dashed line that you use in between the oxigen and the nitrogen is useless and wrong. if you would like to point out the bond make it longer and wider and use for the charges the circled ones: if not we seems that there are 3 minus around
-Scheme 2
nice used of the circled charges, but it is missing a crucial step, when you use a dashed lines is a bond or a link that is apprearing or braking...and then you don't show how it will end, in the next part there is another dashed line etc.
choose: or you explain the process entirely or you do as the "propagation" part, few arrows and the product. When you like to talk about a mechanism is nice to complete all the steps (moreover you have space)
The last three dashed bond are totally wrong.
-line 220
put a citation about CD that explain the affirmation you did.
-line 238
these difficult names required a very clean and codified theory:
you wrote this:
(poly(SBL-alt-GPE/MPE)-b)
I would write all the names as this
poly(SBL-alt-GPE)-block-poly(MPE)
as reported in https://doi.org/10.1351/PAC-REP-12-03-05
Author Response
Comments to the Author
You have well presented a Ring-Opening Copolymerization of γ-Butyroselenolactone and Epoxy Monomer; usign a variety of epoxidic derivates and supporting your thesis with analysis and data. My main concern is more about the way you communicate the results, the aim of a paper is being clear, not interpretable; the results you show have to arrive to the reader unambiguously. You have a lot of data and you did all the analysis neccessary to be published, but there are errors that make me ask myself if you have done it on purpose: i'd like to give you this main take home message.
Q1: The abstract has to be clear: line 17-19 The influence of the catalyst, solvent, and reaction temperature on the reaction efficiency was examined. And then kinetic study was investigated under optimized condition. "And" at the beginning is an error, a full stop is different from a semicolon
A1: Thank you very much for your comment to improve the quality of the manuscript. The abstract was revised carefully to be clear.
Q2: line 21 you meant MALDI?
A2: Yes, the abbreviation was corrected.
Q3: line 54-55 you said:<<However, these monomers were needed to prepare by specific reaction process and it is difficulty to derivatization of these monomer to given more possibilities for polymer modification.>> I really don't know what you want to say...the preposition "to" has to be borne by a subject and a verb.
A3: Thank you for your comment to further improve the quality of the manuscript. It was revised to make the introduction more logical.
Q4: line 69 you want to say that Zhang et al. did something similar, but you end up by pasting a phrase at the end of your introduction: it sounds weird. Moreover is it the same person as one of the coauthor of the paper? if yes, said and reshape the frase, why you wanted to mention that?
A4: Thank you for your comment to further improve the quality of the manuscript. It was revised to make this part more logical.
Q5: line 88-89 add some information more about the MALDI: the matrix you used, the power of the laser etc., how you analized, practially, the polymers, it is important if someone want to replicate the analysis. If you were inspired by a pubblished protocol cite it.
A5: Thank you for your comment to further improve the quality of the manuscript. The detail information about MALDI-TOF was added into the main text.
Q6: line 99-100 in the same phrase cm-1 is appearing in two different forms: being homogeneous.
A6: The mistake was corrected
Q7: line 108 you miss a letter in the title
A7: Thanks, this mistake was corrected.
Q8: Scheme 1 you have to cite what is "Cat." up to the first arrow. The reader should be able to understand the scheme without reading the text.
A8: Thank you for your comment to further improve the quality of the manuscript. The Scheme was redesigned. TBAB was added onto the equation as catalyst.
Q9: line 172 MALDI-TOF you already declare the acronym...but MS no!, you have to declare all the acronym (even if obvious).
A9: Thank you for your comment to further improve the quality of the manuscript. all of the acronym were checked. MS was declared in Section 2.2 Measurement.
Q10: line 186 I'm not an expert in a MALDI, but as is not a HR-MS I would use less decimals Resolution and mass accuracy in matrix-assisted laser desorption ionization- time-of-flight https://doi.org/10.1016/S1044-0305(98)00069-5
A10: Thank you for your suggestion to further improve the quality of the manuscript. Although the data was provided directly from MALDI, two decimal significant digits was proper in MALDI-TOF MS.
Q11: Figure 1 the dashed line that you use in between the oxygen and the nitrogen is useless and wrong. if you would like to point out the bond make it longer and wider and use for the charges the circled ones: if not we seems that there are 3 minus around
A11: Thank you very much for your comment to improve the quality of the manuscript. The structure of polymer was corrected in Figures 1 and 2.
Q12: Scheme 2 nice used of the circled charges, but it is missing a crucial step, when you use a dashed lines is a bond or a link that is appearing or braking...and then you don't show how it will end, in the next part there is another dashed line etc. choose: or you explain the process entirely or you do as the "propagation" part, few arrows and the product. When you like to talk about a mechanism is nice to complete all the steps (moreover you have space). The last three dashed bond are totally wrong.
A12: Thank you very much for your comment to improve the quality of the manuscript. The quality of intermediate in the mechanism was improved as shown in Scheme 3.
Q13: line 220 put a citation about CD that explain the affirmation you did.
A13: Thank you very much for your comment to improve the quality of the manuscript. The reference was added to the main text (Angew. Chem. Int. Ed. 2020, 59, 2-11).
Q14: line 238 these difficult names required a very clean and codified theory: you wrote this: (poly(SBL-alt-GPE/MPE)-b) I would write all the names as this poly(SBL-alt-GPE)-block-poly(MPE) as reported in https://doi.org/10.1351/PAC-REP-12-03-05
A14: Thank you very much for your suggestion to improve the quality of the manuscript. We used poly(SBL-alt-GPE)-b-poly(SBL-alt-MPE) instead of the former name.

Round 2
Reviewer 1 Report
I note with satisfaction that the authors respected my comments and recommendations.
The quality of the manuscript was imiproved significantly.
I recommend to accept the paper in the present form.